# Mitochondrial Genome Variation in Polish Elite Athletes

**DOI:** 10.3390/ijms241612992

**Published:** 2023-08-20

**Authors:** Agnieszka Piotrowska-Nowak, Krzysztof Safranow, Jakub G. Adamczyk, Ireneusz Sołtyszewski, Paweł Cięszczyk, Katarzyna Tońska, Cezary Żekanowski, Beata Borzemska

**Affiliations:** 1Institute of Genetics and Biotechnology, Faculty of Biology, University of Warsaw, 5a Pawińskiego Street, 02-106 Warszawa, Poland; a.piotrowska1@uw.edu.pl (A.P.-N.); k.tonska@uw.edu.pl (K.T.); 2Department of Biochemistry and Medical Chemistry, Pomeranian Medical University, Powstańców Wielkopolskich 72, 70-111 Szczecin, Poland; chrissaf@mp.pl; 3Department of Theory of Sport, Józef Piłsudski University of Physical Education, Marymoncka 34 Street, 00-968 Warszawa, Poland; jakub.adamczyk@awf.edu.pl; 4Department of Forensic Medicine, Medical University of Warsaw, Oczki 1 Street, 02-007 Warszawa, Poland; ireneusz.soltyszewski@wum.edu.pl; 5Faculty of Physical Education, Gdansk University of Physical Education and Sport, Górskiego 1 Street, 80-336 Gdansk, Poland; pawel.cieszczyk@awf.gda.pl (P.C.); c.zekanowski@imdik.pan.pl (C.Ż.); 6Department of Neurogenetics and Functional Genomics, Mossakowski Medical Research Institute, Polish Academy of Sciences, Pawińskiego 5 Street, 02-106 Warszawa, Poland

**Keywords:** athletes, mitochondrial DNA, copy number, haplogroups, heteroplasmy, variants

## Abstract

Energy efficiency is one of the fundamental athletic performance-affecting features of the cell and the organism as a whole. Mitochondrial DNA (mtDNA) variants and haplogroups have been linked to the successful practice of various sports, but despite numerous studies, understanding of the correlation is far from being comprehensive. In this study, the mtDNA sequence and copy number were determined for 99 outstanding Polish male athletes performing in power (n = 52) or endurance sports (n = 47) and 100 controls. The distribution of haplogroups, single nucleotide variant association, heteroplasmy, and mtDNA copy number were analyzed in the blood and saliva. We found no correlation between any haplogroup, single nucleotide variant, especially rare or non-synonymous ones, and athletic performance. Interestingly, heteroplasmy was less frequent in the study group, especially in endurance athletes. We observed a lower mtDNA copy number in both power and endurance athletes compared to controls. This could result from an inactivity of compensatory mechanisms activated by disadvantageous variants present in the general population and indicates a favorable genetic makeup of the athletes. The results emphasize a need for a more comprehensive analysis of the involvement of the mitochondrial genome in physical performance, combining nucleotide and copy number analysis in the context of nuclear gene variants.

## 1. Introduction

The overall heritability of sports-related phenotypes is estimated at ca. 66% [1,2]. It is widely accepted that an elite athletic performance is strongly dependent on the genomic background. To date, numerous association studies have pinpointed more than 200 variants in the nuclear as well as the mitochondrial genome generally associated in particular populations with various phenotypic traits related to sports performance [3].

Energy efficiency is one of the essential features of the cell and the organism affecting sports performance. The ATP is provided by creatine/creatine phosphate and glycolytic systems in the cytoplasm and by oxidative phosphorylation (OXPHOS) in mitochondria. Aerobic endurance performance requires prolonged muscle activity and a continuous supply of ATP produced mostly by OXPHOS. Power and sprint sports rely mostly on anaerobic glycolysis. 

Although the majority of mitochondrial proteins are encoded by the nuclear DNA, thirteen subunits of the OXPHOS system are encoded by the mitochondrial genome (mtDNA). The cross-talk of the mitochondrial and nuclear genomes plays a crucial role in maintaining the energy balance. More than three decades ago, an association between variants in mtDNA and aerobic capacity was shown for the first time in family studies [4,5]. Since then, numerous reports have shown associations of mtDNA variants and haplogroups with elite athletic performance [6,7]. The results were generally population-specific, not fully reproducible, and stronger for endurance sports, which mainly rely on OXPHOS efficiency.

Mounting evidence indicates a role of the variable mtDNA copy number in diseases and aging [8]. However, little is known about possible mtDNA copy number correlations with athletic performance, especially as endurance-related variables are associated not only with success in sports but also with good health and low mortality [9,10]. Moreover, novel data suggest that the proportion of mutated mtDNA copies is not the only determinant of disease and that the absolute copy number of mtDNA matters. It is also known that the mtDNA copy number reflecting the abundance of mitochondria in a cell may change under different energy demands, including physiological conditions related to aerobic training [11,12]. 

The aim of our study was to assess whether specific mtDNA sequence variation, especially haplogroup distribution, rare variants, heteroplasmy, and copy number, are associated with elite athlete performance in the Polish population. 

## 2. Results

To determine the contribution of mtDNA genetic factors to outstanding athletic performance, we performed a detailed analysis of mitochondrial genome variation in 99 male elite athletes from the Polish population and compared it with 100 sex- and age-match sedentary controls (C). The athletes were divided into two groups according to the metabolic processes dominating their sports discipline: anaerobic-dominant sports, later called power sports (P, n = 52), and aerobic-dominant sports, later referred to as endurance sports (E, n = 47). 

No pathogenic single nucleotide variants, according to the MITOMAP database [13], were found in any sample. 

The mitochondrial genome was covered by an average sequencing depth of 5653 ± 1690 x with 520,269 ± 147,709 reads per sample.

### 2.1. mtDNA Haplogroup Distribution

Thirteen different mitochondrial haplogroups were identified. Their frequencies are presented in Figure 1. The haplogroup distribution showed no statistically significant differences between the studied groups (*p* = 0.41). European haplogroup H was the most common in all groups.

### 2.2. SNV Association Analysis

We compared the frequency of single nucleotide variants (SNV) among all groups analyzed. We selected for further analysis 117 variants present (>5%) in at least five subjects of both groups combined (athletes + controls) and conducted a statistical analysis of all possible comparisons between the study groups (P + E vs. C; P vs. E; P vs. C; E vs. C).

Homoplasmic variants (≥95% vs. <5%) m.194C>T, m.8994G>A, m.11947A>G, and m.12414T>C were underrepresented in athletes (P + E) compared with the control group (*p* = 0.022 for each of this four variants). Variant m.709G>A was underrepresented in P compared with E athletes (*p* = 0.016) and controls (*p* = 0.017). Variants m.930G>A, m.11812A>G, m.14233A>G, m.16294C>T, and m.16296C>T were underrepresented in P compared with controls (0.02 < *p* < 0.05). For details, see Table 1. 

We compared the proportion of heteroplasmic vs. homoplasmic subjects for three variants (m.520_523dup, m.522_523dup, m.16093T>C), which were heteroplasmic in at least five subjects and found no significant differences between groups (*p* > 0.6).

Finally, for 16 variants found in a heteroplasmic state in at least one subject, we compared “variant dose”, i.e., the percentage amount treated as a quantitative variable between groups using a non-parametric Mann–Whitney test. The only significant difference was observed for the m.16296C>T variant, which was less frequent in the P group compared with controls (*p* = 0.034), consistently with the underrepresentation of homoplasmic subjects with this variant in the P group. No statistical significant association was observed for the endurance athletes. 

False discovery rate (FDR) used to account for 413 (≥95% vs. <5% amount) + 8 (heteroplasmic vs. homoplasmic) + 54 (variant dose) = 475 tests performed in SNV association analysis showed q-value = 0.73 for each of the ten variants underrepresented in athletes or in their P subgroup. This value indicates with a very high probability that, in fact, all the associations found above are false positives.

### 2.3. Rare mtDNA Variants

The distribution of rare mtDNA variants (as listed in the MITOMAP database ≤0.5%) showed no difference between any of the groups studied when assessed by variant burden (Figure 2). We also compared the number of participants by frequency of variants that have 0, 1, 2, or more rare mtDNA sequence changes. Here, power and endurance athletes did not differ significantly from controls or from each other (Appendix A.

### 2.4. Heteroplasmic mtDNA Variants

On average, the athletes had significantly less heteroplasmic variation than the controls (all athletes 0.33 ± 0.59, controls 0.65 ± 0.88, mean ± SD, *p* = 0.0079, Appendix A). The effect was much more pronounced for the endurance athletes (0.28 ± 0.50, mean ± SD, *p* = 0.01) than for power ones, where only a tendency was observed (0.38 ± 0.66, mean ± SD, *p* = 0.07) in relation to controls. The difference between the power and endurance groups was not significant (*p* = 0.51). Additionally, the percentage of individuals with heteroplasmic variants showed a markedly different distribution between the study groups (Figure 3, Appendix A). The athletes generally, and the endurance athletes especially, were less likely to carry heteroplasmic variants when compared to control subjects who tended to carry multiple heteroplasmic variants in mtDNA. Notably, as many as 69% power athletes and 74% endurance ones, and 56% controls were homoplasmic (Figure 3, Appendix A).

### 2.5. Non-Synonymous and Synonymous mtDNA Variants

The proportion of non-synonymous (NS) and synonymous (S) variants in protein-coding genes is commonly accepted as a good indicator of the action of natural selection on the genetic variation analyzed. We, therefore, determined the distribution of non-synonymous and synonymous variants in all protein-coding mtDNA regions of the groups studied. No significant differences between any of the groups were found (Figure 4). Additionally, the distribution of subjects with 0, 1, 2 or more non-synonymous or synonymous variants did not differ significantly between the groups (Appendix A). 

### 2.6. mtDNA Copy Number

We determined the absolute mtDNA copy number in blood and saliva from the ratio of *MT-ND1* and the nuclear *B2M* gene and using copy number reference standards (Figure 5). The athletes had fewer mtDNA copies than did the controls, both in blood (71 ± 22 vs. 83 ± 17, mean ± SD, *p* = 0.00028) and in saliva (94 ± 38 vs. 127 ± 64, mean ± SD, *p* = 0.034). Further analysis showed that the different mtDNA copy number distribution observed in blood was primarily influenced by endurance athletes (67 ± 23, mean ± SD, *p* = 0.00005), whereas in saliva from power athletes (92 ± 38, mean ± SD, *p* = 0.024). The mtDNA copy number did not differ substantially either in blood (*p* = 0.079) or saliva (*p* = 0.35) between P and E athletes (Figure 5). The MT-ND1/B2M measures were greater in saliva than in blood in both controls (*p* = 0.000009) and all athletes together (*p* = 0.0033) influenced by endurance athletes (*p* = 0.012), but not by power athletes (*p* = 0.19). mtDNA copy number ranged from 27 to 142 in blood and 40 to 343 in saliva.

Table 2 summarizes the results of the verification of all possible hypotheses.

## 3. Discussion

Mitochondria are the powerhouses of cells, but they also regulate many other cellular processes, including steroid biosynthesis, gluconeogenesis, ion homeostasis, cellular calcium signaling, and programmed cell death, confirming them as central hubs of metabolism. Therefore, the particular efficiency of these organelles may underlie not only personal well-being but also outstanding athletic performance. To verify this assumption at the genetic level, we analyzed mitochondrial haplogroup distribution, frequency of rare mtDNA variants, proportions of non-synonymous and synonymous variants, heteroplasmy, and mtDNA copy number in a group of elite power and endurance athletes compared with matched sedentary controls. 

As expected, no pathogenic mtDNA variants were detected in the study group or in controls. Although the exact frequency of pathogenic mtDNA variants in Poland is not known, some of the variants may be present in the general population at a heteroplasmy level below the detection threshold (e.g., m.3243A>G) [14] and at the same time, variants with low penetrance (such as m.11778G>A responsible for Leber hereditary optic neuropathy) may be present in healthy subjects. Based on their prevalence in other countries and the gnomAD database, the probability of finding a pathogenic mtDNA variant in a healthy person seems to be relatively low, ranging from 1/5000 to 1/250 [15,16].

The rate of accumulation of mtDNA variants is relatively high due to its solely maternal inheritance, a high mutation rate resulting from redox processes, a lack of efficient DNA protection mechanisms, and virtually no recombination. Some variants are in linkage disequilibrium and are inherited together as haplotypes (haplogroups) reflecting the population history. Therefore, different mitochondrial haplogroups are specific to geographic regions, e.g., haplogroups H, V, U, K, T, J, W, I, and X dominate in Europe [17,18]. The regional specificity of mtDNA lineages suggests that mtDNA variation has facilitated human survival in different climatic zones. According to the Wallace hypothesis, haplogroups associated with OXPHOS coupling/uncoupling affect mitochondrial efficiency and thus predispose activities, including athletic performance [18]. Indeed, some data indicate that mtDNA haplogroups and subhaplogroups associated with mitochondrial uncoupling are more common in speed athletes, e.g., K and J2 in Finnish speed athletes, or are less common in endurance male athletes, e.g., K in Polish ones [6,19]. However, in general, the haplogroups do not correlate strongly with an athletic status or markers of good trainability (e.g., maximal oxygen uptake, VO2max). 

Here, we did not find a statistically significant differences in haplogroup distribution between athletes and controls or between endurance and power athletes. Thus, we could not confirm our previous results [6]. This may be due to the smaller study group resulting from the application of strict achievement criteria and the inclusion of only the most outstanding Polish athletes. The influence of mitochondrial haplogroups on athletic performance seems to be very subtle to be adequately evaluated and requires a large and as homogeneous group as possible (in respect of the type of muscle work, energy system, sports class, ethnic origin, age, etc.), nearly impossible to assemble. Not only the association between the elite athletic status and haplogroups, but also that with common mtDNA variants is population-specific and difficult to identify. 

Several studies have demonstrated the influence of mtDNA variants on aerobic performance and training response. However, the results were not reproducible [20,21,22,23]. Additionally, the statistically significant variants found in our study do not reflect the earlier results. Harvey et al. (2020) pointed to the benefits of simultaneous analysis of the nuclear and mitochondrial genomes, highlighting the role of the mitochondrion-nucleus cross-talk, difficult to identify in association studies [21]. Most of the known variants in mtDNA do not affect mitochondrial functioning in a straightforward manner. However, some rare mtDNA variants, in most cases belonging to particular subhaplogroups, have functional consequences and have been associated with athletic performance [24]. Some of them can potentially tighten the OXPHOS coupling, others could cause uncoupling. Recently, Kiskilla et al. (2019) suggested that endurance athletes harbor an excess of rare mtDNA variants that may be beneficial for OXPHOS, whereas sprinters could tolerate mtDNA variants with a disease-causing potential, adversely affecting OXPHOS [25].

An accumulation of rare mtDNA variants, especially non-synonymous ones, is considered a risk factor for mitochondrial dysfunction [26]. Therefore, athletes could be expected to harbor a lower number of such variants with less negative impact on mitochondrial performance. However, our study does not support this hypothesis: the frequency of rare mtDNA variants and non-synonymous ones showed no statistically significant differences between the groups studied.

Heteroplasmy is common in the case of pathogenic mtDNA variants but is also observed for benign and neutral variants and seems to be quite prevalent in the general population. The mechanisms shaping its origins and dynamics are not fully understood, making its role in the context of fitness and sports achievements unclear. It has been suggested that heteroplasmy could be disadvantageous, even in the case of neutral variants. In organisms with biparental mtDNA inheritance, hybrids with two distant mtDNA types often develop mitochondrial pathologies [27]. 

We observed that athletes, especially the endurance ones, are less heteroplasmic than the controls. Homoplasmy was detected in 69% of power athletes and 74% of endurance athletes but only in 56% of the control. If indeed heteroplasmy is an unfavorable condition, the homoplasmic individuals should have better-functioning mitochondria and, therefore better sports performance. Simultaneously, our study showed a lower mtDNA copy number in athletes compared with age-matched sedentary controls. The variability of mtDNA copy number is widely discussed in the literature in the context of pathology and aging, but not sports achievements. 

The presence of pathogenic mtDNA variant seems to be positively correlated with the mtDNA copy number. In primary mitochondrial diseases, an upregulation of mtDNA copy number, usually due to an upregulated mitochondrial biogenesis, is regarded as a compensatory mechanism to sustain cellular bioenergetics. A high mtDNA copy number often correlates with a decreased disease severity or even an incomplete penetrance. This effect is believed to be due to both an increased ratio of wild-type mtDNA to the mutant form in heteroplasmic individuals and also to the elevation of the absolute mtDNA copy number [8].

The lower mtDNA copy number we found in athletes compared with age-matched sedentary controls could have resulted from the absence of a compensatory reaction to harmful variants absent in the athletes and, by inference, indicates a favorable set of genetic variants in athletes.

On the other hand, Hagman et al. (2021) studied various aging markers, including mtDNA copy number, in young and elderly elite athletes compared with age-matched controls [28]. In contrast to our results, they found that young elite soccer players had a higher mtDNA copy number in white blood cells compared to young controls. This may be due to the different energy requirements of team sports. 

It should be noted that the number of mtDNA copies depends on the specific physiological situation of the cell and organism. For example, in cancer, it strongly depends on the type of tumor and can show a positive correlation with both increased and decreased severity of disease symptoms [8]. 

A pronounced reduction in the mtDNA copy number associated with higher mortality, poor health, and cognitive decline is observed in people in their 50s and older [29,30]. A reduced mtDNA copy number has also been reported across different brain regions in patients affected by neurodegeneration [31,32]. A decreased mitochondrial efficiency with aging and in age-dependent diseases is believed to be the primary cause of the decline in vital functions rather than its consequence. Nilsson and Tarnopolsky (2019) have suggested that in the general population, aerobic exercise training of older adults may partially reverse mitochondrial dysfunction by increasing the mtDNA copy number and mitochondrial volume, transcript and protein expression, and OXPHOS efficiency [33]. One should note, however, that chronic training is a stressor that results in improvements in mitochondrial volume, structure, and functioning only in organs/tissues experiencing an increased energy demand [34].

Additionally, in contrast to the reports discussed above, Hagman et al. (2021) found that the mtDNA copy number was slightly positively correlated with age and modified by prior training status [28].

What is more, studies on animal models showed that high mtDNA copy number can, in fact, be detrimental to mitochondrial functioning by leading to an altered nucleoid architecture, which in turn impairs mtDNA transcription and thereby causes a progressive OXPHOS dysfunction [35]. It could, therefore, be speculated that maintaining an adequate mtDNA copy number is crucial for cellular homeostasis. However, the mechanisms of mtDNA copy number regulation are far from being fully understood. Moreover, the mtDNA copy number is variable in different tissues, reflecting differences in the physiological activity of different tissues depending on environmental requirements. This may be the reason for the differences we observed in mtDNA copy numbers in cells isolated from saliva and blood. This further shows that the effects of the mtDNA copy number on the physiological parameters of health and disease can vary depending on multiple variables.

Our work is one of the first to highlight the significance of mtDNA copy number variation and heteroplasmy in the context of athletic performance. This seems to be particularly important since endurance-related variables are not only associated with athletic success but also with good health and low mortality. We believe that our results will not only contribute to the field of sports genetics but will also shed light on mitochondrial function. However, we are aware that further, comprehensive studies using much larger homogeneous groups from different populations and taking into account the context of the variants of the nuclear genes involved in mitochondrial biogenesis and function are needed to decipher the role of the genetic components modulating predisposition to elite athletic performance.

## 4. Materials and Methods

### 4.1. Materials

The study group comprised 99 male athletes (mean age 24.3 ± 5.8) and represented the following performance levels: (1) top national level (National Elite Class) with at least one medal at Polish national championships, and (2) international level (International Elite Class) with participation (finals, medal positions) at the European/World Championships or Olympic Games. Using the above criteria, 52 athletes were classified as International Elite Class and 47 as National Elite Class (Table 3). The achievements of the International Elite Class are described in Table 4.

The control group consisted of 100 age- and sex-matched sedentary subjects (male, mean age 22.5 ± 1.7), unengaged in competitive sports or in any formal, supervised training, and without medical history of any diseases affecting the heart or blood vessels. All subjects were unrelated Caucasians of Polish origin.

### 4.2. DNA Isolation

DNA was isolated from peripheral blood leukocytes using the standard salting-out procedure and from saliva using the Oragene DNA Self-Collection Kit and Prep IT L2P Purification Kit (DNA Genotek Inc., Stittsville, ON, Canada) according to the manufacturer’s instructions. 

DNA samples were collected from peripheral blood (n = 28 power athletes, n = 42 endurance athletes, and n = 70 controls) or saliva (n = 24 power athletes, n = 5 endurance athletes, and n = 30 controls). 

### 4.3. Structure of the Study

Mitochondrial genome sequence analysis was performed for 99 athletes and 97 of 100 control subjects (three samples failed to amplify the entire mtDNA). The mtDNA copy number was determined for 97 of 99 athletes and 98 of 100 control subjects. Four samples yielded unreliable results due to the presence of the m.3915G>A variant at a position corresponding to the 3′ end of one of the PCR primers used for quantification and were therefore excluded from further analysis. More details can be found in Appendix A.

### 4.4. mtDNA Sequencing

The mitochondrial genome sequence was determined using next-generation sequencing (NGS) as described previously [36,37]. In brief, full mtDNA was amplified as a single fragment or two overlapping ones using long-range PCR. The PCR products were purified and quantified before proceeding to DNA library preparation for sequencing using the Illumina Nextera XT protocol and sequencing on a MiSeq instrument. The sequencing data were processed using CLC Genomics Workbench v11 software (CLC bio, Qiagen, Hilden, Germany). A bioinformatic analysis comprised quality control of the sequencing reads, mapping to the human mtDNA reference sequence (rCRS, GenBank sequence NC_012920), variant detection, annotation, and evaluation based on the strategy described in detail previously [36]. Variants detected in mtDNA regions found to be challenging to sequence (such as homopolymer stretches) and thus of low reliability were excluded from further analysis. Those regions were m.303_315, m.567_573, m.956_965, m.5895_5899, and m.16180_16193. In this study, heteroplasmic mitochondrial variants were called for sequence changes if supported by 5–95% of NGS reads covering a given position in the mitochondrial genome. If a variant was present in at least 95% of the reads obtained, it was considered homoplasmic. The mtDNA variants of heteroplasmy below 5% were not investigated here.

Based on the full set of sequence variants detected, mtDNA haplogroups were assigned with Haplogrep, a mtDNA haplogroup classification tool [38]. The variant population frequency was obtained from the MITOMAP database [13]. The frequency cutoff value for rare mtDNA variants was set at ≤0.5%.

Detailed data on the mtDNA features investigated in this study for each subject is given in Appendix A.

### 4.5. Determination of mtDNA Copy Number

We developed a method for mtDNA copy number determination using an absolute quantification approach based on the copy number standards and real-time PCR using SYBR Green assay on a Roche LightCycler 480 instrument. A detailed description of the methodology used can be found in Appendix A.

### 4.6. Statistical Analysis

Statistical analysis was performed for the following comparisons: (1) all athletes versus the control group (P + E vs. C), (2) power athletes versus the control group (P vs. C), (3) endurance athletes versus the control group (E vs. C), and (4) power athletes versus endurance athletes (P vs. E). The distribution of mitochondrial haplogroups was compared between groups using the chi-square test. Mann–Whitney test was used to evaluate differences in the number of rare, non-synonymous, synonymous, heteroplasmic variants, proportion of non-synonymous variants (NS/(NS + S)), and mtDNA copy number between groups. All statistical hypotheses (comparisons of one parameter between groups) were grouped into six categories (haplogroup distribution, SNV associations, rare variants, heteroplasmic variants, synonymous and non-synonymous variants, mtDNA copy number), and p-value was calculated for each hypothesis. Then, to account for multiple testing, FDR correction was applied, and q-value was calculated for each category. To account for multiple categories, Bonferroni-like correction was applied: the typical threshold of q-value < 0.2, corresponding to >80% probability that a positive test represents a true difference, was divided by the number of categories (6) to yield q < 0.033 as the final criterion of statistical significance.

Detailed results of all statistical analyses are given in Appendix A.

## 5. Conclusions

Our results indicate a lack of an unequivocal association of individual mtDNA haplogroups or common and rare mtDNA variants with elite athletic status in the analyzed groups. On the other hand, the elite athletes, especially the endurance ones, harbor more homoplasmic variants as compared to controls and have a lower mtDNA copy number. We speculate that the low abundance of heteroplasmic variants may improve mitochondrial functioning and, therefore, favor elite athletic performance. The relatively low mtDNA copy number likely reflects a lack of a compensatory reaction to unfavorable mtDNA variants absent in elite athletes. However, the exact mechanisms underlying heteroplasmy disadvantage are elusive. 

Our results indicate that the mitochondrial genome should be studied comprehensively, combining nucleotide sequence and copy number analysis and taking into account the context of the variants of the nuclear genes involved in the biogenesis and functioning of mitochondria, especially under the conditions of sports training.

## Figures and Tables

**Figure 1 ijms-24-12992-f001:**
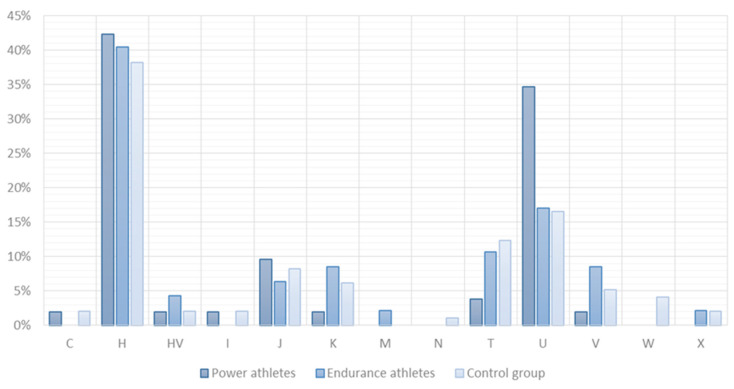
mtDNA haplogroup distribution among power athletes, endurance athletes, and controls.

**Figure 2 ijms-24-12992-f002:**
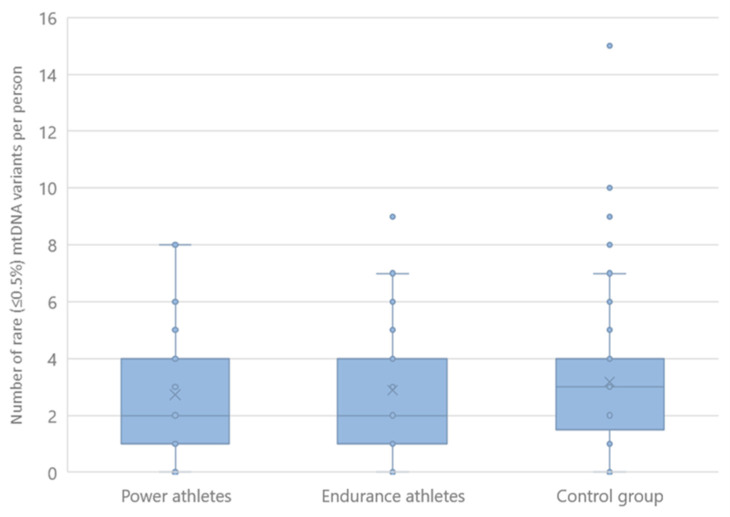
Distribution of rare (≤0.5%) mtDNA variants in athletes and controls. Mean values are marked with a cross (all athletes 2.81 ± 2.26, power athletes 2.73 ± 1.92, endurance athletes 2.89 ± 2.61, control group 3.18 ± 2.40, mean ± SD). The differences were not statistically significant (P + E vs. C *p* = 0.17, P vs. C *p* = 0.27, E vs. C *p* = 0.26, P vs. E *p* = 0.76, Mann–Whitney test).

**Figure 3 ijms-24-12992-f003:**
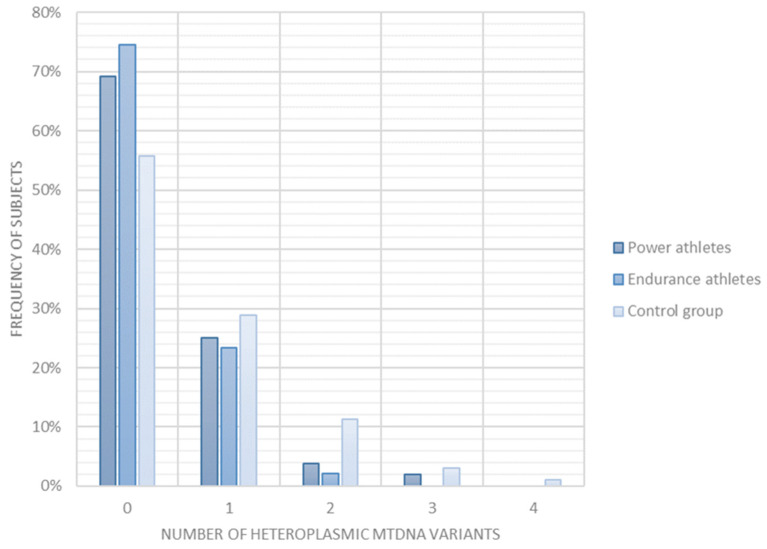
Frequency of power and endurance athletes and controls with 0, 1, 2 or more heteroplasmic mtDNA variants (P + E vs. C *p* = 0.0079, P vs. C *p* = 0.07, E vs. C *p* = 0.01, P vs. E *p* = 0.51, Mann–Whitney test).

**Figure 4 ijms-24-12992-f004:**
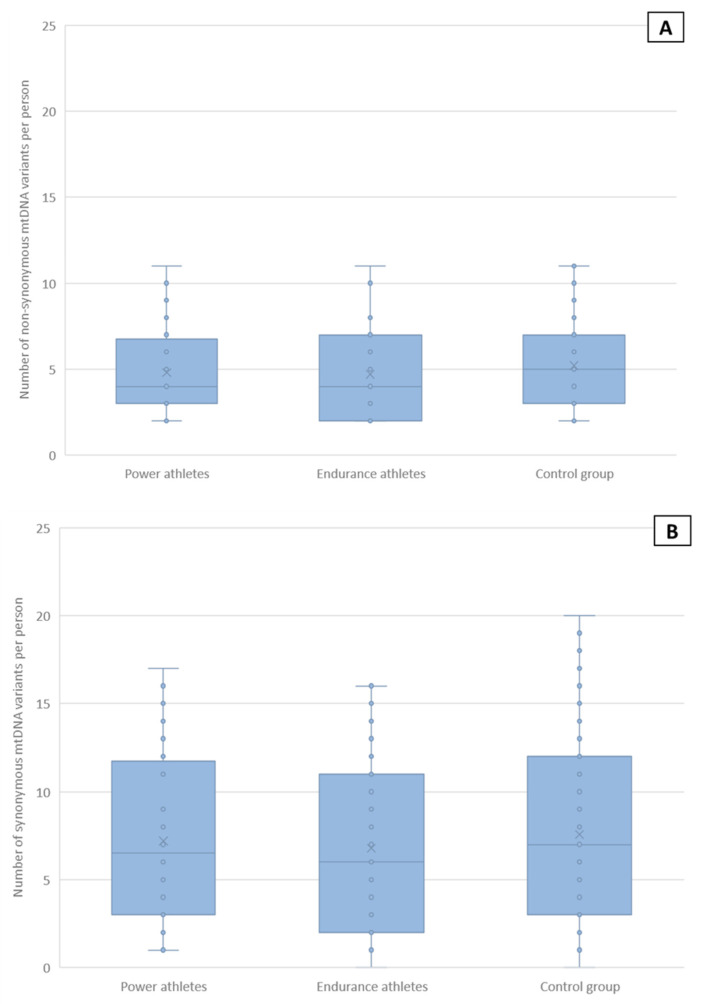
Distribution of synonymous and non-synonymous mtDNA variants in study groups. (**A**), non-synonymous (NS) variants, (**B**), synonymous (S) variants, (**C**), proportions of non-synonymous variants to all variants (NS/NS + S). Mean values are marked with a cross. No statistically significant differences were observed (NS: P + E vs. C *p* = 0.14, P vs. C *p* = 0.31, E vs. C *p* = 0.17, P vs. E *p* = 0.49; S: P + E vs. C *p* = 0.47, P vs. C *p* = 0.79, E vs. C *p* = 0.35, P vs. E *p* = 0.51; NS/(NS + S): P + E vs. C *p* = 0.98, P vs. C *p* = 0.75, E vs. C *p* = 0.78, P vs. E *p* = 0.63, Mann–Whitney test).

**Figure 5 ijms-24-12992-f005:**
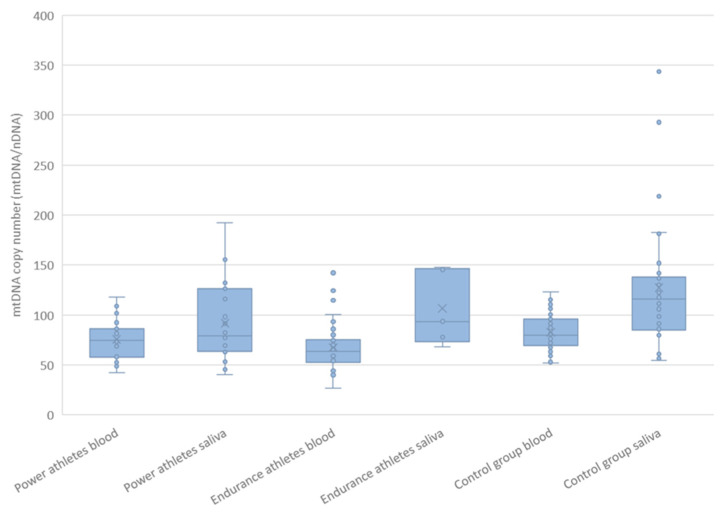
Distribution of mtDNA copy number in blood (nP = 27, nE = 42, nC = 68) and saliva (nP = 23, nE = 5, nC = 30) in athletes and controls. Mean values are marked with a cross (blood: all athletes 71 ± 22, power athletes 75 ± 20, endurance athletes 67 ± 23, controls 83 ± 17; saliva: all athletes 94 ± 38, power athletes 92 ± 38, endurance athletes 107 ± 38, controls 127 ± 64, mean ± SD). Statistical significance of differences: blood P + E vs. C *p* = 0.00028, E vs. C *p* = 0.00005 (P vs. C *p* = 0.13, P vs. E *p* = 0.079); saliva P + E vs. C *p* = 0.034, P vs. C *p* = 0.024 (E vs. C *p* = 0.64, P vs. E *p* = 0.35), Mann–Whitney test.

**Table 1 ijms-24-12992-t001:** Frequency of statistically significant mtDNA SNVs (*p* < 0.05 without correction for multiple tests).

mtDNA Variant	Power Athletes (n = 52)	Endurance Athletes (n = 47)	Control Group (n = 97)
m.194C>T	0%	0%	5% $,^
m.709G>A	4% $,#	19%	18%
m.930G>A	0% #	6%	9%
m.8994G>A	0%	0%	5% $,^
m.11812A>G	2% #	6%	11%
m.11947A>G	0%	0%	5% $,^
m.12414T>C	0%	0%	5% $,^
m.14233A>G	2% #	6%	11%
m.16294C>T	4% #	11%	14%
m.16296C>T	0% #	6%	8%

# *p* < 0.05 vs. control group; $ *p* < 0.05 vs. endurance athletes; ^ *p* < 0.05 vs. power athletes.

**Table 2 ijms-24-12992-t002:** Verification of hypotheses after FDR correction for multiple comparisons.

Category of Hypotheses	Number of Hypotheses Tested	Associations Considered True Positive *	*p*-Value	q-Value
Haplogroup distribution	1	-	0.41	0.41
mtDNA SNV associations	475	-	>0.015	>0.73
Rare mtDNA variants	4	-	>0.17	>0.35
Heteroplasmic mtDNA variants	4	number of heteroplasmic mtDNA variants is lower in athletes than in controls	0.0079	0.027
		number of heteroplasmic mtDNA variants is lower in endurance athletes than in controls	0.014	0.027
Synonymous and non-synonymous mtDNA variants	12	-	>0.14	>0.86
mtDNA copy number	12	mtDNA copy number in blood is lower in endurance athletes than in controls	0.00005	0.0003
		mtDNA copy number in blood is lower in athletes than in controls	0.00028	0.0011
		mtDNA copy number is higher in saliva than in blood in controls	0.000009	0.00011
		mtDNA copy number is higher in saliva than in blood in athletes	0.0033	0.0099
		mtDNA copy number is higher in saliva than in blood in endurance athletes	0.012	0.028

* q-value < 0.033.

**Table 3 ijms-24-12992-t003:** Characteristics of groups of athletes by discipline and performance level (International or National Elite Class).

	Sport	n	International Elite Class	National Elite Class
Power sports	Athletics, sprints (100–400 m) and throws	27	25	2
	Weightlifting	23	0	23
	Powerlifting	1	0	1
	Swimming (sprint)	1	1	0
	Total	52	26	26
Endurance sports	Athletics, long-distance runs (3000 m—marathon)	10	9	1
	Cross-country skiing	9	4	5
	Swimming (400–1500 m)	22	8	14
	Triathlon	3	2	1
	Rowing	2	2	0
	Speed skating	1	1	0
	Total	47	26	21

**Table 4 ijms-24-12992-t004:** Sporting achievements of International Elite athletes (as of 5 November 2022).

	Olympic Games			World Championships			European Championships		
	Participant, Only	Finalist, Only	Medalists	Participant, Only	Finalist, Only	Medalists	Participant, Only	Finalist, Only	Medalist
Power sports	16	9	4	21	18	11	26	19	14
Endurance sports	9	4	2	10	8	5	24	10	6
Total	25	13	6	31	26	16	50	29	20

Note: An athlete could take part in more than one type of event, so the numbers in rows do not sum up.

## Data Availability

Data supporting the findings of this study are available from the corresponding author on request.

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
