# Peer review of "Mitochondrial Genome Variation in Polish Elite Athletes"

_ijms, 2023, doi:10.3390/ijms241612992_

Round 1
Reviewer 1 Report
The study is well-planned, seriously done, interesting and original. However, there are some concerns that need to be addressed before the manuscript could be considered for publication in the MDPI International Journal of Molecular Sciences.
1. Abbreviations should be defined at first mention and used consistently thereafter, e.g. line 98: SNV; Line 118: FDR method, etc.
2. Line 82: It is not clear what gender the subjects in the control group were. Were there only male too?
3. The quality of all Figures is blurry.
4. Figure 1. It is not clear what mtDNA haplogroups (C, H, HV, I, J, K, M, N, T etc.) are identified and shown in the figure? What is the mtDNA haplogroups in the world (across different nations)? Please describe it both in the text and in the figure.
5. Figure 3. It is not clear what the mtDNA variants are – 0, 1, 2, 3, 4
6. Line 162 - 168: Please provide options of non-synonymous and synonymous variants in protein coding genes, i.e. gene name, variant nomenclature.
7. Line 182: Genes names should be rewritten in Italics.
8. There are some typing mistakes, e.g. lines 53.
9. Discussion: please provide accurate or additional information about strengths and limitation of the study.
10. Materials and Methods
a. Line 342 and Line 359: Again, it is unclear what the controls was? Are sedentary subjects being male? The authors indicate "The control group consisted of 100 age" - what is it?
Questions for authors:
· Why was mtDNA isolated from blood and saliva? is it not enough to isolate mtDNA only from blood samples? Can the difference in mtDNA sequence be determined in blood and saliva of the same subject?
· What mtDNA variants are found in the study? are there differences between SNV and non-synonymous or synonymous variants - are they not the same variants? The authors do not specify the mtDNA variants nomenclature.
Author Response
The study is well-planned, seriously done, interesting and original. However, there are some concerns that need to be addressed before the manuscript could be considered for publication in the MDPI International Journal of Molecular Sciences.
We are very grateful for the opinions and valuable comments that helped us improve the manuscript.
- Abbreviations should be defined at first mention and used consistently thereafter, e.g. line 98: SNV; Line 118: FDR method, etc.
We have followed the reviewer's instructions throughout the text.
As a result, we moved the excerpt (lines 342-345) to the beginning of the Results Section to clarify the abbreviations: P (power athletes) and E (endurance athletes).
2.Line 82: It is not clear what gender the subjects in the control group were. Were there only male too?
Yes, they were also male. Missing information has been added to the text. - The quality of all Figuresis blurry.
We have prepared and resubmitted a file for all figures in a sufficiently high resolution. - Figure 1. It is not clear what mtDNA haplogroups (C, H, HV, I, J, K, M, N, T etc.) are identified and shown in the figure? What is the mtDNA haplogroups in the world (across different nations)? Please describe it both in the text and in the figure.
Figure 1 shows the frequency of all mtDNA haplogroups (described as major macrohaplogroups or top-level haplogroups) identified in all subjects studied in this work. The data are shown for the three groups studied in this work, that is power athletes, endurance athletes, and controls. Detailed data on the specific mtDNA haplogroup for each subject can be found in Supplementary Materials 1, as highlighted in the Materials and Methods chapter (lines 400-401).
The distribution of mtDNA haplogroup varies among populations in the world – a simplified mtDNA tree can be found in the MITOMAP database (https://www.mitomap.org/foswiki/pub/MITOMAP/WebHome/simple-tree-mitomap-2019.pdf). It is therefore crucial to compare ethically matched groups in population-based studies. This was done in our study. The distribution of mtDNA haplogroup described in this paper is similar to those reported in other studies of similar European populations. However, comparing the mtDNA haplogroup distribution between different nations was not the aim of our study.
- Figure 3. It is not clear what the mtDNA variants are – 0, 1, 2, 3, 4
Figure 3 shows the frequency of subjects in the study groups (power athletes, endurance athletes, and controls) with 0, 1, 2, 3, or 4 heteroplasmic variants identified in their mtDNA. The numbers 0-4 in Figure 3 reflect how many different heteroplasmic variants were found in one’s mtDNA. - Line 162 - 168:Please provide options of non-synonymous and synonymous variants in protein coding genes, i.e. gene name, variant nomenclature.
Detailed information on all identified variants can be found in Supplementary Materials 1, as highlighted in the Materials and Methods chapter (lines 400-401). In this study, we focused on collective analysis of mtDNA variation: non-synonymous variants (or synonymous variants) from all protein-coding mtDNA regions were counted for each group and compared cumulatively.
- Line 182:Genes names should be rewritten in Italics.
The correction has been made as suggested by the Reviewer. - There are some typing mistakes, e.g. lines 53.
The typing errors have been corrected. - Discussion: please provide accurate or additional information about strengths and limitation of the study.
A paragraph has been added at the end of the Discussion section:
Our work is one of the first to highlight the significance of mtDNA copy number variation and heteroplasmy in the context of athletic performance. This seems to be particularly important since endurance-related variables are not only associated with athletic success but also with good health and low mortality. We believe that our results will not only contribute to the field of sports genetics but will also shed light on mitochondrial function. However, we are aware that further, comprehensive studies, using much larger homogeneous groups from different populations and taking into account the context of the variants of the nuclear genes involved in mitochondrial biogenesis and function are needed to decipher the role of the genetic components modulating predisposition to elite athletic performance.
- Materials and Methods
Line 342 and Line 359: Again, it is unclear what the controls was? Are sedentary subjects being male? The authors indicate "The control group consisted of 100 age" - what is it?
The sentence has been corrected.
Questions for authors:
- Why was mtDNA isolated from blood and saliva? is it not enough to isolate mtDNA only from blood samples? Can the difference in mtDNA sequence be determined in blood and saliva of the same subject?
This issue is related to subject consent– not all subjects consented to blood sample collection and preferred to provide saliva samples, especially during intensive training and competition. To correct for different tissue sources, a similar approach was used for control subjects. For each subject, one source material (blood or saliva) was collected. Therefore, we are not able to compare the mtDNA variation between two different tissues from the same subject.
- What mtDNA variants are found in the study? are there differences between SNV and non-synonymous or synonymous variants - are they not the same variants? The authors do not specify the mtDNA variants nomenclature.
Detailed information on all identified variants can be found in Supplementary Materials 1, as highlighted in Materials and Methods section (lines 400-401).
SNV – Single Nucleotide Variant, is a general category, while non-synonymous/synonymous variant reflects the effect on the primary structure of the protein. SNV association analysis was also performed in this study and the results are described in section 2.2.
Reviewer 2 Report
Congratulations to the authors for a work with a good sample and a good scientific relevance
Author Response
We are very grateful for the Reviewer’s positive opinion.
Reviewer 3 Report
The article titled "Mitochondrial Genome Variation in Polish Elite Athletes" delves into the intriguing realm of genetic variations within the mitochondria of elite athletes in Poland. While the topic is undoubtedly captivating, the introduction requires substantial revision to meet the standards of the authors' submission rules. It lacks a clear and concise description of the study's objectives, methodology, and the significance of the findings.
One glaring issue with the article is the peculiar placement of the methods section at the end of the article. Typically, methods are an essential component of any research paper, and their appropriate placement should be near the beginning of the article. Placing the methods section towards the end hinders readers from understanding the study's design, data collection, and analysis procedures early on, thus undermining the clarity and validity of the findings.
By revising the introduction to clearly state the study's objectives, moving the methods section to an appropriate location, and enhancing the analysis, discussion, and conclusion sections, the article would be better equipped to captivate readers and contribute meaningfully to sports research. Furthermore, addressing structural issues and refining the introduction and results chapter will enhance the overall readability and impact of the research.
Author Response
We would like to point out that we followed the manuscript structure recommended by The Editors. In order to prepare the manuscript for IJMS more efficiently, we used the Microsoft Word template suggested in the Instructions for Authors. Therefore, the layout of the sections in the manuscript is fully consistent with the editorial requirements.
We do believe that our manuscript has been prepared with due diligence, and detailed information has been included in appropriate sections in the most transparent, consistent, and readable form.
According to Reviewer’s suggestion, we have slightly modified the sentence indicating the research objectives in the Introduction section (lines 78-79).
Reviewer 4 Report
Review of the article "Mitochondrial genome variation in Polish elite athletes" First of all, I would like to state that the article is in line with the latest trends in research and scientific discoveries in sports. Combining phenotypic traits and looking for a cause-and-effect relationship with genetics as a whole is an appropriate and enlightening method. Recent genetic research includes not only polymorphisms of selected genes, but also haplotypes, epigenetics, methylation, and gene expression.
In the case of athletes, the task is further complicated by the fact that certain characteristics of "athletic excellence" are not only polygenic, but also multifactorial.
The study of the mitochondrial genome in Polish elite athletes is an excellent thematic and methodological choice, because it is well known that mitochondrial DNA (mtDNA) variants and haplogroups have been associated with success in various sports, but despite numerous studies, the understanding of the correlation is far from complete.
In their study, the authors determined the mtDNA sequence and copy number of 99 outstanding Polish male athletes involved in power sports (n=52) or endurance sports (n=47) and 100 controls.
Considering the research methodology and the phenotypically narrowed test group, the sample size is large and fully satisfactory.
No correlation was found between any haplogroup, single nucleotide variant, especially rare or nonsynonymous one, and athletic performance. Interestingly, heteroplasma was less frequent in the study group, especially in endurance athletes. Lower mtDNA copy number was found in both power and endurance athletes compared to controls. This could result from an inactivity of compensatory mechanisms activated by disadvantageous variants present in the general population and indicates a favourable genetic make-up of the athletes. The results emphasize a need for a more comprehensive analysis of the involvement of the mitochondrial genome in physical performance, combining nucleotide and copy number analysis in the context of nuclear gene variants. The manuscript is written in good, understandable language with sufficient description of the methodology. The authors conducted the discussion correctly.
I am happy to recommend the manuscript for printing, and I encourage the authors to continue their research, especially in the area of the mitochondrial genome.
Author Response

(The authors gave the same response as above.)
